# Is ChatGPT Transforming Academics' Writing Style?

## Abstract

Based on one million arXiv papers submitted from May 2018 to January 2024, we assess the textual density of ChatGPT's writing style in their abstracts through a statistical analysis of word frequency changes. Our model is calibrated and validated on a mixture of real abstracts and ChatGPT-modified abstracts (simulated data) after a careful noise analysis. The words used for estimation are not fixed but adaptive, including those with decreasing frequency. We find that large language models (LLMs), represented by ChatGPT, are having an increasing impact on arXiv abstracts, especially in the field of computer science, where the fraction of LLM-style abstracts is estimated to be approximately 35%, if we take the responses of GPT-3.5 to one simple prompt, "revise the following sentences", as a baseline. Papers from other disciplines have been relatively less impacted. We conclude with an analysis of both positive and negative aspects of the penetration of LLMs into academics' writing style.

## 1 Introduction

Since ChatGPT (Chat Generative Pre-trained Transformer) was released on November 30, 2022, large language models (LLMs) have become widely available and begun to affect many aspects of our lives. Many papers have explored the advantages and disadvantages of LLMs (Kasneci et al., 2023), along with their capabilities (Srivastava et al., 2022). Although they can increase productivity and may help scientific discovery (Noy & Zhang, 2023; AI4Science & Quantum, 2023), the potential risks of using LLMs in academia cannot be ignored (Lund et al., 2023) – for example, generating incorrect references (Walters & Wilder, 2023) or unintended plagiarism. In this paper, we are concerned with whether LLMs, represented by ChatGPT, are transforming a specific activity, namely, academic writing.

For example, LLMs can generate the abstract of a paper directly when provided with a suitable prompt (Luo et al., 2023). And studies have shown that identifying such abstracts is not easy, even if they remain unedited by humans (Gao et al., 2023; Cheng et al., 2023). Within the broad field of academic writing and publishing, we chose the abstracts of articles as the focus of this work, as they have a relatively uniform format across disciplines, are supposed to condense an entire research article and thus are often highly polished, and can be considered short articles of pure text, not involving pictures nor tables.

We want to analyze the fingerprints of LLMs on scientific abstracts as a function of time in order to tease out a statistical signature, rather than a binary classification. There are many methods for detecting LLM-generated text. But the detection of a mixture of human and machine-generated text is usually much harder (Krishna et al., 2024; Zhang et al., 2024), which we think is more common in real scenarios.

While there is already a corpus of current research on using ChatGPT in academia (Casal & Kessler, 2023; Lingard et al., 2023; Fergus et al., 2023; Lund et al., 2023), to our knowledge only a handful of works have attempted to quantify its impact on the whole academic community. When the first version of this paper was being completed, two papers appeared that addressed related questions: one focuses on AI conferences peer reviews (Liang et al., 2024a), the other analyzes scientific papers (Liang et al., 2024b). They claim that the usage of LLMs is evident in AI conference reviews and scientific writings, especially in computer science papers.

However, there have been some issues in previous research. For instance, in the work of Liang et al. (2024b) the estimated fraction of LLM-modified sentences is least 2% before ChatGPT, indicating that there is a

bias in the estimate or that the definition of the estimate needs to be modified. In addition, there are no articles discussing how to automatically select words to estimate the impact of LLMs. Existing research on this topic also lacks rigorous modeling and analysis, which is also our focus in this paper. Once the reliability of a single analysis is assured, the comprehensive analysis can be more convincing.

Furthermore, we think that the estimation of the usage or impact of ChatGPT should be a relative value, as people using ChatGPT can achieve entirely different outcomes, for instance, by employing different prompts. Therefore, we use the more neutral term "LLM impact" instead of "proportion" in our estimation. Because the estimates in this paper are based on ChatGPT simulations, the effect is also called "ChatGPT impact".

The main contribution of this paper is to propose a framework for estimating the impact of LLMs on academic writing. This framework is based on word frequency analysis and simulation, supported by a theoretical analysis of noise and variability. For instance, different words would yield different estimation results, and more adaptive approaches are used in our framework, as well as considering words with decreasing frequency.

## 2 Data

**arXiv dataset**   The metadata of arXiv papers are provided by Kaggle (arXiv.org submitters, 2024). Because the abstracts in this dataset are updated when authors submit changes, we used the first version in 2024 (version 161) as well as the last version before the ChatGPT era (version 105). Our observations and analysis are based on one million arXiv articles submitted from May 2018 to January 2024.

**English word frequency**   Google Ngram dataset is chosen for comparison and reference (Michel et al., 2011). Specifically, we used the freely available mirrors on Kaggle (`http://kaggle.com/datasets/wheelercode/english-word-frequency-list`) covering word frequencies from the 1800s to 2019 as established from Google Books.

## 3 Changes in Word Frequency

### 3.1 Observations

We approach the problem by analyzing how the frequency of words changes after ChatGPT has been deployed. The frequency of some non-specialized words (such as "significant", "effectively", "crucial" and others) starts to skyrocket in early 2023, as presented in Figure 1, where 1 million abstracts are divided into 100 uneven time-periods, each encompassing 10,000 abstracts. As a larger the sample size improves the accuracy of any estimate, we used the same number of articles in each period rather than the same time interval to keep the error of our estimates constant, providing the same quality of observation and estimation.

It is very suggestive that the frequencies of all those words has begun to grow very significantly at the same time. Another striking example is the frequency change of the words "are" and "is". The counts in 10,000 abstracts of these two words were quite stable before 2023. However, the frequency of these two terms has dropped by more than 10% in 2023.

These examples, anecdotal as they are, may represent the tip of the iceberg of a wider and growing phenomenon: the rapid increase in the usage of ChatGPT or other LLMs. The rise and fall in frequency of specific technical nouns may well be related to the changing popularity of certain research topics, but that a research trend is responsible for the change in usage of adjectives and/or non-technical terms appears implausible – even less so for extremely common words such as "is" and "are". In order to investigate further, we turned to simulation.

### 3.2 LLM Simulations

The emergence of other LLMs is also inspired or influenced by ChatGPT, and we also assume that other LLMs have similar but not identical word preferences to ChatGPT. The most recent articles we processed were submitted in January 2024, when we expect other LLMs to be less prevalent due to the market predominance of ChatGPT. Therefore, we used ChatGPT for simulations.

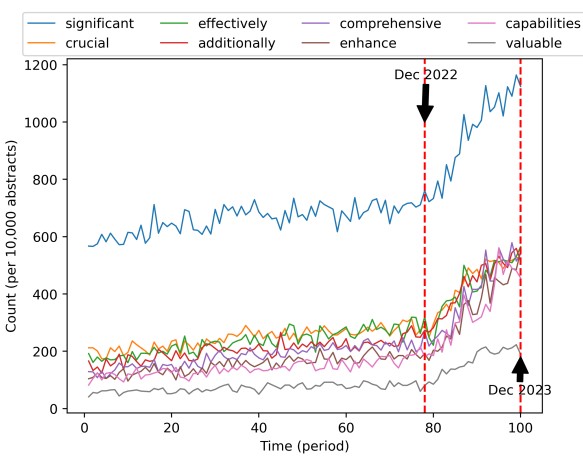 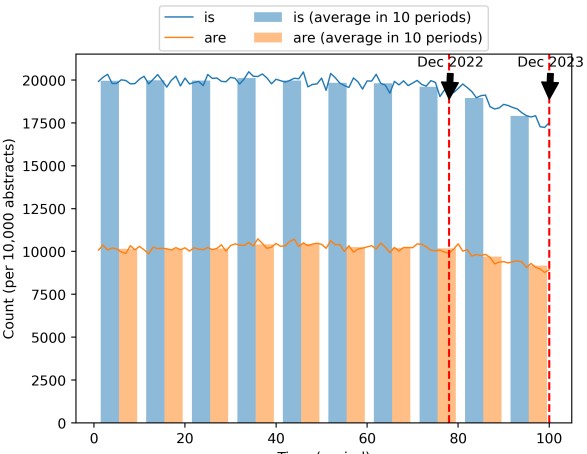

Figure 1: Word frequency changes in abstracts. The vertical red dashed line demarcates the first time period after ChatGPT's release.

What prompts might be used in real life is unknowable, and we think simple prompts could better reflect the inherent word preferences of ChatGPT, as complex prompts may bring more human interference. We adopt the neutral prompt, and the parameters are listed in Section F of the Appendix:

**P1**: *"Revise the following sentences:"*

In order to be more specific about the impact of LLMs on articles from different disciplines, we analyses arXiv abstracts from different categories separately, with particular attention to the four categories with the highest number of articles: *cs* (computer science), *math* (mathematics), *astro* (astrophysics), and *cond-mat* (condensed matter). The one million arXiv articles were divided into 20 time periods in the analysis in order to increase the number of articles per period, and thus reduce the estimation error. For example, in the *cs* category, each period has more than 10,000 articles, while the other three categories each have at least 3,000. The identifier numbers of the first and last arXiv articles corresponding to each period are given in section B of the Appendix.

We also select a portion of the abstracts published before the emergence of ChatGPT and simulate them using GPT-3.5, which should be the most widely used LLM before 2024. For example, the simulation results of 10,000 abstracts in period 14 (April 2022 to July 2022) for the 4 arXiv categories with the most articles are shown in Table 1. We found that ChatGPT processing alters the frequencies of many words, including the words "is", "are", and "significant" that we mentioned earlier.

Table 1: Word frequency (per abstract) before and after ChatGPT processing (prompt **P1**).

| words | category | before | after | change rate |
|---|---|---|---|---|
| is, are | cs | 2.01, 1.00 | 1.73, 0.83 | -14%, -17% |
| is, are | math | 1.78, 0.74 | 1.61, 0.71 | -9%, -5% |
| is, are | astro | 2.13, 1.39 | 1.90, 1.25 | -11%, -1% |
| is, are | cond-mat | 2.00, 0.92 | 1.68, 0.80 | -16%, -13% |
| significant | cs | 0.09 | 0.18 | 99% |
| significant | math | 0.01 | 0.03 | 308% |
| significant | astro | 0.17 | 0.26 | 53% |
| significant | cond-mat | 0.07 | 0.18 | 171% |

Although GPT-3.5 may not have the same word preferences as other LLMs or indeed subsequent ChatGPT versions, this observation corroborates the hypothesis, formulated earlier, that the drop in the frequency of the words "is", "are" observed in real abstracts in 2023 may have been caused by ChatGPT. Combined with Figure 8 in the Appendix showing the correlation between changes in simulated and real data, we speculate that ChatGPT is one of the important reasons, possibly even the main reason, for the recent word frequency change in abstracts.

### 3.3 Modelling

Our next step is to model the LLM impact or ChatGPT impact, as well as estimating the impact based on real data and simulations.

We start with a simple model, ignoring noise and variability here. In order to minimize the influence of the research topic, different words should be used for estimation for different paper categories. Additionally, it is important to consider not only words that increase in frequency, but also those that decrease in frequency.

Suppose that the frequency of word $i$ for abstracts in subject category $j$ changes from $f_{ij}^*$ to $\tilde{f}_{ij}^*$ after being processed by ChatGPT, when it is used as a means to polish and improve the abstract (if not to fully generate it). The corresponding word change rate is defined as

$$\bar{r}_{ij} = \frac{\tilde{f}_{ij}^* - f_{ij}^*}{f_{ij}^*} = \frac{\tilde{f}_{ij}^*}{f_{ij}^*} - 1 \,. \tag{1}$$

Suppose that $f_{ij}^d(t)$ is the word frequency for word $i$ in category $j$ at time period $t$, this can be written as:

$$f_{ij}^d(t) = (1 - \eta_j(t))f_{ij}^*(t) + \eta_j(t)f_{ij}^*(t)(\bar{r}_{ij} + 1) = f_{ij}^*(t) + \eta_j(t)f_{ij}^*(t)\bar{r}_{ij} \tag{2}$$

where $\eta_j(t)$ denotes the proportion of abstracts in category $j$ affected by LLMs, and $f_{ij}^*(t)$ represents the original evolution in word frequency without LLMs.

## 4 Impact Estimation

### 4.1 Simple Estimation

Our next goal is to figure out how to better estimate $\eta_j(t)$. A simple and strategy would be to first determine the values of the other variables in Eq. (2), which will allow it to become a linear regression problem.

Although there is no direct way to investigate ChatGPT's word preference, we can ask ChatGPT to polish or rewrite real, pre-2023 abstracts, and use the resulting simulation data to calculate the estimated frequency change rate $\hat{r}_{ij}$ of word $i$ in category $j$:

$$\hat{r}_{ij} = \frac{\tilde{q}_{ij}^d - q_{ij}^d}{q_{ij}^d} = \frac{\tilde{q}_{ij}^d}{q_{ij}^d} - 1 \tag{3}$$

where $q_{ij}^d$ represents the word frequency of real abstracts in the dataset and $\tilde{q}_{ij}^d$ means the frequency after ChatGPT processing.

We cannot know the true value of $f_{ij}^*(t)$ in the LLM era, but we can replace it with the estimation $\hat{f}_{ij}^*(t)$ based on the word frequency before LLMs were introduced. As our objective is to identify the words that LLM "likes" (or "dislikes") to use compared to academic researchers on average, we assume that the frequencies of these words should remain stable without LLM, i.e., we take the average of the pre-ChatGPT periods (before $T_0$) as follows:

$$f_{ij}^*(t) = \frac{1}{\#\{t \leq T_0\}} \sum_{t \leq T_0} f_{ij}^d(t), \text{if } t > T_0 \tag{4}$$

where $f_{ij}^d$ represents the frequency of the word $i$ in the category $j$ observed in real data. For example, we used the first 10 periods before ChatGPT was introduced, to calculate $f_{ij}^*(t)$, as they weren't influenced by ChatGPT, which means $T_0 = 10$ and $\#\{t \leq T_0\} = 10$ in Eq. (4).

Since $\bar{r}_{ij}$ and $f_{ij}^*(t)$ can be approximated with Eq. (3) and Eq. (4), we have one estimate of $\eta_j(t)$ for a given word $i$. If we consider a set of words $I_j$, the estimate of the LLM impact based on Eq. (2) can be obtained using Ordinary Least Squares (OLS):

$$\hat{\eta}_j(t) = \frac{\sum_{i \in I_j} (f_{ij}^d(t) - f_{ij}^*(t)) f_{ij}^*(t) \hat{r}_{ij}}{\sum_{i \in I_j} (f_{ij}^*(t) \hat{r}_{ij}^2)} \, . \tag{5}$$

Figure 2 presents the estimation results based on different groups of words (listed in Section F.2 of the appendix), where the simulation results of 20,000 abstracts in period 13 are used to estimate $r_{ij}$.

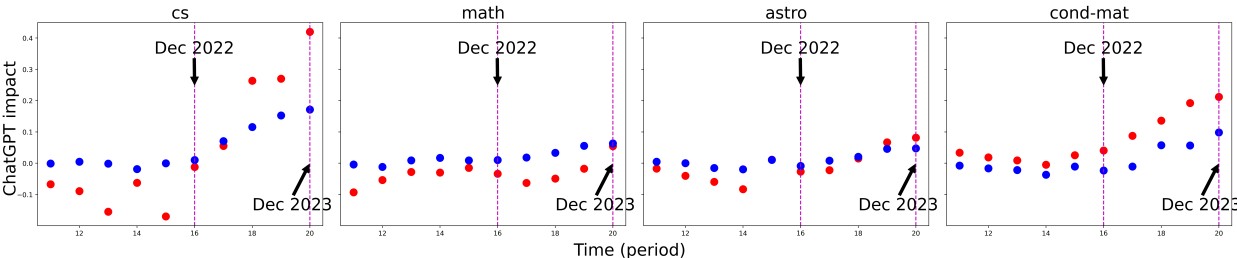

Figure 2: Examples of estimating the ChatGPT impact based on different word groups. Each point represents an estimate, and each color corresponds to estimations derived from the same set of words.

Estimates based on different words will yield different results. Which words should we choose for the estimation?

## 4.2 Noise Analysis

In many data analysis applications, more data points translates into better estimates. But in our case, the effect of noise is different for each data point (word), as discussed in Section G.1 of the appendix. Considering the computational complexity, the exhaustive method is impractical for selecting words. Thus, we intend to choose words according to some criteria.

Our starting point is noise and variability, such as randomness within the LLM, uncertainty in the evolution of word usage without LLM, and epistemic uncertainty in how users actually prompt LLMs. The noise terms might be modeled in many different ways, for example,

$$f_{ij}^d = f_{ij}^* + \delta_{ij}(f_{ij}^*) \tag{6}$$

where $\delta_{ij}(\cdot)$ represents noise and word usage variability which are not directly related to the internal parameters of the LLM.

We assume that the noise in the "real" data and in the simulations due to LLM processing can be represented as $\epsilon_{ij}(\cdot)$ and $\epsilon_{ij}^s(\cdot)$. Then Eq. (1) and Eq. (3) are related by

$$\frac{\tilde{f}_{ij}^* - \epsilon_{ij}(f_{ij}^*) - f_{ij}^*}{f_{ij}^*} = \frac{\tilde{q}_{ij}^d - \epsilon_{ij}^s(q_{ij}^d) - q_{ij}^d}{q_{ij}^d} \, . \tag{7}$$

After taking into account the impact of LLMs, we split the word frequency term $f_{ij}^d(t)$ into two parts, $f_{ij}^{\delta,\eta}(t)$ and $f_{ij}^{\delta,1-\eta}(t)$, each with their corresponding noise term:

$$f_{ij}^{\delta,\eta}(t) = \eta_j(t) f_{ij}^*(t) + \delta_{ij}(\eta_j(t) f_{ij}^*(t)) \tag{8}$$

$$f_{ij}^{\delta,1-\eta}(t) = (1 - \eta_j(t)) f_{ij}^*(t) + \delta_{ij}((1 - \eta_j(t)) f_{ij}^*(t)) \, . \tag{9}$$

Similar to the idea of Eq. (7), we can define the influence of noise in "real" and simulated data on the estimate $\hat{r}_{ij}$ as the following variable:

$$\epsilon_{ij}^{\eta}(q, f, t) = \frac{\epsilon_{ij}(f_{ij}^{\delta,\eta}(t))}{f_{ij}^{\delta,\eta}(t)} - \frac{\epsilon_{ij}^{s}(q_{ij}^{d})}{q_{ij}^{d}} \,. \tag{10}$$

In this case, the equation corresponding to Eq. (2) is

$$\begin{aligned} f_{ij}^{d}(t) &= (1 - \eta_j(t))f_{ij}^{*}(t) + \delta_{ij}((1 - \eta_j(t))f_{ij}^{*}(t)) + f_{ij}^{\delta,\eta}(t)(\hat{r}_{ij} + 1 + \epsilon_{ij}^{\eta}(q, f, t)) \\ &= f_{ij}^{\delta,1-\eta}(t) + f_{ij}^{\delta,\eta}(t)(\hat{r}_{ij} + 1 + \epsilon_{ij}^{\eta}(q, f, t)) \,. \end{aligned} \tag{11}$$

Then, Eq. (11) – representing the difference in word frequency before and after LLM processing – can be rewritten as

$$f_{ij}^{d}(t) - f_{ij}^{*}(t) = \eta_j(t)f_{ij}^{*}(t)\hat{r}_{ij} + g_{ij}(t) + \xi_{ij}(t) \tag{12}$$

with

$$g_{ij}(t) = \eta_j(t)f_{ij}^{*}(t)\epsilon_{ij}^{\eta}(q, f, t) \tag{13}$$

$$\xi_{ij}(t) = (\hat{r}_{ij} + 1 + \epsilon_{ij}^{\eta}(q, f, t))\delta_{ij}(\eta_j(t)f_{ij}^{*}(t)) + \delta_{ij}'((1 - \eta_j(t))f_{ij}^{*}(t)) \,. \tag{14}$$

where $\delta_{ij}'(\cdot)$ follows the same distribution as $\delta_{ij}(\cdot)$.

It should be noted that $g_{ij}(t)$ includes only LLM-related noise $\epsilon_{ij}(\cdot)$ and $\epsilon_{ij}^{s}(\cdot)$, however $\xi_{ij}(t)$ contains $\delta_{ij}(\cdot)$ and $\delta_{ij}'(\cdot)$ that are unrelated to LLM.

In order to estimate $\eta_j(t)$, we can use the quadratic loss function

$$L_{j,t}(\eta_j) = \frac{1}{n_j} \sum_{i \in I_j} (f_{ij}^{d}(t) - f_{ij}^{*}(t) - \eta_j(t)f_{ij}^{*}(t)\hat{r}_{ij})^2 = \frac{1}{n_j} \sum_{i \in I_j} (g_{ij}(t) + \xi_{ij}(t))^2 \tag{15}$$

where $n_j$ represent the number of words in $I_j$.

If we ignore the dependency of $g_{ij}(t)$ and $\xi_{ij}(t)$ on $\eta_j(t)$, the estimate of LLM impact could be given like before, i.e., Eq. (5). However, since $g_{ij}(t)$ also depends on $\eta_j(t)$ and $\xi_{ij}$ contains $\eta_j(t)$ as described in Eq. (13) and Eq. (14), we need to make additional assumptions to progress further. Our motivation is to improve the estimation in Eq. (5) based on assumptions and analysis of the noise terms.

### 4.3 Case Study

**Case 1:** if the effect of $\eta_j(t)$ on $\xi_{ij}(t)$ can be ignored compared to other terms, e.g., the following simple scenario,

$$\mathrm{Var}[\delta_{ij}(\eta_j(t)f_{ij}^{*}(t))] \ll \eta_j(t)f_{ij}^{*}(t)\mathrm{Var}[\epsilon_{ij}^{\eta}(q, f, t)] \tag{16}$$

One can also derive the approximation below:

$$f_{ij}^{\delta,\eta}(t) \approx \eta_j(t)f_{ij}^{*}(t) + \delta_{ij}(*) \tag{17}$$

where $\delta_{ij}(*)$ is a random variable with zero mean and variance much smaller than $\eta_j(t)f_{ij}^{*}(t)$, and its value after being divided by $\eta_j(t)$ is negligible relative to $f_{ij}^{*}(t)$.

For the sake of convenience in representation, we define the following terms:

$$h_{ij}(t) = f_{ij}^{d}(t) - f_{ij}^{*}(t) \tag{18}$$

$$x_{ij}(t) = f_{ij}^{*}(t)\hat{r}_{ij} \,. \tag{19}$$

Therefore, the loss function under the assumption is:

$$L_{j,t,g}(\eta_j) = \frac{1}{n_j} \sum_{i \in I_j} (h_{ij}(t) - \eta_j(t)x_{ij}(t) - g_{ij}(t))^2 = \frac{1}{n_j} \sum_{i \in I_j} \xi_{ij}^2(t) \,. \tag{20}$$

Thus,

$$\frac{\partial L_{j,t,g}(\eta_j)}{\partial \eta_j} = \frac{2}{n_j} \sum_{i \in I_j} \left( \eta_j(t) x_{ij}^2(t) - h_{ij}(t) x_{ij}(t) \right) + \frac{2}{n_j} \sum_{i \in I_j} x_{ij}(t) g_{ij}(t)$$
$$- \frac{2}{n_j} \sum_{i \in I_j} \frac{\partial g_{ij}(t)}{\partial \eta_j(t)} \left( h_{ij}(t) - \eta_j(t) x_{ij}(t) - g_{ij}(t) \right) \tag{21}$$

If we require a minimum by setting $\frac{\partial L_{j,t,g}(\eta_j)}{\partial \eta_j} = 0$, we obtain a new estimate $\hat{\eta}_j^g(t)$, which is equal to the OLS $\hat{\eta}_j(t)$ in Eq. (5) corrected for bias and noise,

$$(\hat{\eta}_j^g(t) - \hat{\eta}_j(t)) \sum_{i \in I_j} x_{ij}^2(t) = \sum_{i \in I_j} \frac{\partial g_{ij}(t)}{\partial \eta_j(t)} \left( h_{ij}(t) - \eta_j(t) x_{ij}(t) \right)$$
$$- \sum_{i \in I_j} x_{ij}(t) g_{ij}(t) - \sum_{i \in I_j} g_{ij}(t) \frac{\partial g_{ij}(t)}{\partial \eta_j(t)} . \tag{22}$$

But without knowing the distribution of $\epsilon_{ij}(\cdot)$ and $\epsilon_{ij}^s(\cdot)$, we have no way of estimating the value of this bias, so we assume that $\epsilon_{ij}(f_{ij}) \sim \mathcal{N}(0, f_{ij}\sigma_{ij,\epsilon}^2)$ and $\epsilon_{ij}^s(f_{ij}) \sim \mathcal{N}(0, f_{ij}\sigma_{ij,\epsilon}^2)$, e.g., $\epsilon_{ij}(1) \sim \mathcal{N}(0, \sigma_{ij,\epsilon}^2)$. Then, by combining Eq. (10) and Equation Eq. (13), we can obtain the following relationship:

$$g_{ij}(t) = \eta_j(t) f_{ij}^*(t) \left( \frac{\epsilon_{ij}(f_{ij}^{\delta,\eta}(t))}{f_{ij}^{\delta,\eta}(t)} - \frac{\epsilon_{ij}^s(q_{ij}^d)}{q_{ij}^d} \right) = \frac{\eta_j(t) f_{ij}^*(t) \epsilon_{ij}(1)}{\sqrt{\eta_j(t) f_{ij}^*(t) + \delta_{ij}(*)}} - \frac{\eta_j(t) f_{ij}^*(t) \epsilon_{ij}^s(1)}{\sqrt{q_{ij}^d}} . \tag{23}$$

Therefore, all terms on the right-hand side of Eq. (22) are zero-mean noise, except for the last one:

$$g_{ij}(t) \frac{\partial g_{ij}(t)}{\partial \eta_j(t)} = g_{ij}(t) \frac{f_{ij}^*(t)(\eta_j(t) f_{ij}^*(t) + 2\delta_{ij}(*))\epsilon_{ij}(1)}{2(\eta_j(t) f_{ij}^*(t) + \delta_{ij}(*))^{\frac{3}{2}}} - g_{ij}(t) \frac{f_{ij}^* \epsilon_{ij}^s(1)}{\sqrt{q_{ij}^d}} . \tag{24}$$

Removing the items with zero means, we get

$$\mathrm{E}\left[ g_{ij}(t) \frac{\partial g_{ij}(t)}{\partial \eta_j(t)} \right] = \frac{\eta_j(t)(f_{ij}^*(t))^2(\eta_j(t) f_{ij}^*(t) + 2\delta_{ij}(*))\sigma_{ij,\epsilon}^2}{2(\eta_j(t) f_{ij}^*(t) + \delta_{ij}(*))^2} + \frac{\eta_j(t)(f_{ij}^*(t))^2 \sigma_{ij,\epsilon}^2}{q_{ij}^d} . \tag{25}$$

The bias part is expressed as

$$\hat{\eta}_j(t) - \hat{\eta}_j^g(t) = \frac{\sum_{i \in I_j} \mathrm{E}\left[ g_{ij}(t) \frac{\partial g_{ij}(t)}{\partial \eta_j(t)} \right]}{\sum_{i \in I_j} (f_{ij}^*(t) \hat{r}_{ij})^2} . \tag{26}$$

**Insights:** Some insights can be gained from the results above. As by definition $\eta_j(t) \geq 0$, the estimate $\hat{\eta}_j(t)$ given by Eq. (5) tends to be biased high in our model. The value of $\hat{r}_{ij}$ plays a role in the minimization of bias, as it only appears in the denominator in Eq. (26). Similarly, if the value of $\hat{r}_{ij}$ is similar for different words, then larger values of $q_{ij}^d$ and $f_{ij}^*$ will reduce the bias, as seen from Eq. (25) – therefore, we should consider including preferentially in our analysis words with larger values of $q_{ij}^d$, $f_{ij}^*$ and $|\hat{r}_{ij}|$. Considering that the value of $\eta_j(t)$ affects the bias as well, which is not simply linear, we are led to consider adaptive or iterative criteria for word choice, which will in general depend on the true (and unknown) value of $\eta_j(t)$.

**Case 2:** Gaussian distribution for $\delta_{ij}(f_{ij})$, e.g., $\delta_{ij}(f_{ij}) \sim \mathcal{N}(0, f_{ij}\sigma_{ij}^2)$, inspired by the central limit theorem and justified empirically in the Appendix, Figure 10. As a result,

$$\xi_{ij}(t) = (\hat{r}_{ij} + \epsilon_{ij}^\eta(q, f, t))\delta_{ij}(\eta_j(t) f_{ij}^*(t)) + \delta_{ij}'(f_{ij}^*(t))$$
$$= \sqrt{\eta_j(t) f_{ij}^*(t)}(\hat{r}_{ij} + \epsilon_{ij}^\eta(q, f, t))\delta_{ij}(1) + \sqrt{f_{ij}^*(t)}\delta_{ij}'(1) \tag{27}$$

which gives us similar conclusions: it's better to choose words with higher values of $q_{ij}^d$, $f_{ij}^*$ and $|\hat{r}_{ij}|$ (detailed calculations can be found in Section G.2 of the Appendix).

**Insights:** Finding criteria for selecting the words that are included in the frequency change analysis greatly reduces the computational complexity compared to trying different word combinations. If all combinations of $n$ words are tried, that complexity grows as $O(2^n)$. When we use word choice criteria to select several groups of words, the complexity is reduced to $O(1)$. Our analysis of noise models gives some insights into these criteria, such as $q_{ij}^d$ and $\hat{r}_{ij}$.

## 4.4 Calibration and Test

In order to verify the theoretical and practical validity of our approach, we used calibrations and tests, with ChatGPT-processed abstracts mixed with real abstracts. Considering that the noise in real data is likely highly complex, we did not estimate the variance of $\epsilon_{ij}(\cdot)$. Instead, we used ChatGPT to process additional abstracts (on top of those used to estimate $r_{ij}$), and used the resulting frequencies as calibration for the bias and noise.

As previous analyses have demonstrated, with the goal of reducing bias in estimation, selecting different words likely correspond to different (unknown) ground truth values of $\eta_j(t)$. Therefore, we construct $N$ different sets of abstract data for calibration and test, $D_n$ and $T_{n'}$, with its correspond mixed ratio of ChatGPT-processed abstracts, $\eta_n$ and $\eta'_{n'}$, as

$$(D_n, \eta_n), n \in \{1, 2, \ldots, N\}; \quad (T'_n, \eta'_n), n' \in \{1, 2, \ldots, N'\}. \tag{28}$$

And for one pair of $(D_n, \eta_n)$ and a specific word choice requirement $q_k$ (for example, $q_{ij}^d > 0.1$ and $\dfrac{\hat{r}_{ij} + 1}{\hat{r}_{ij}^2} < \dfrac{0.1 + 1}{0.1^2}$), the efficiency can be defined as

$$e(D_n, \eta_n, q_k) = |\eta_n - \hat{\eta}_n(D_n, q_k)| \tag{29}$$

where $\hat{\eta}_n(D_n, q_k)$ is the estimate of $\eta_n$ using Eq. (5) and the words set $I_j$ can be derived from $q_k$, denoted $I_j(q_k)$.

For a given set of $q_k$ (examples can be found in the Appendix), we are looking for the best one minimizing $e(D_n, \eta_n, q_k)$, denoted $q(D_n, \eta_n)$, which is the calibration part. For the test data $T_{n'}$, the estimate of $\eta_{n'}$ is calculated from Eq. (5) with different $I_j$, based on different $q(D_n, \eta_n)$ obtained in the calibration procedure.

Because of the goal of the calibration, word choice may well actually introduce a new bias to neutralize the original bias, so that the estimate is not necessarily higher in the test results than the ground truth.

To calibrate the choice of set $I_j$, we use different mixing ratios, in proportion to the value of $\eta_j(t)$. In addition, we only consider the 10,000 words with the highest frequency in the Google Ngram dataset. The first 10 periods before ChatGPT was introduced were still used to estimate $f_{ij}^*(t)$. We continue our simulations based on GPT-3.5. As the training data for GPT-3.5 is up to September 2021, abstracts submitted later than this time are considered: 20,000 abstracts in period 13 to estimate $r_{ij}$, 10,000 abstracts in period 12 for calibration, and 10,000 abstracts in period 14 for testing.

We take $\{\eta_n\} = \{0, 0.05, 0.1, \ldots, 0.45, 0.5\}$ and $m = 1$, which means $N = \#\{(D_n, \eta_n)\} = 11$. Then the 11 $I_j$ (with possible repetitions), obtained from mixed data with 11 corresponding $\eta_n$ of period 12, were used for $\eta'_n$ estimation in the test data (period 14). Other parameters can be found in Section F of the Appendix.

The results using the same prompt for generating calibration and test data are shown in Figure 3a, with injected mixed ratio (i.e., ChatGPT impact) $\eta'_n$ from 0 to 0.5. It is clear that when the calibration and test sets are mixed in the same ratio, word combinations that achieve better estimates on the calibration set generally work better on the test set, as well.

Unlike in Figure 3a where we normalized the word frequency by the total number of abstracts, we normalized it by the total number of words for one period in Figure 3b. The trends remain similar, albeit different in detail.

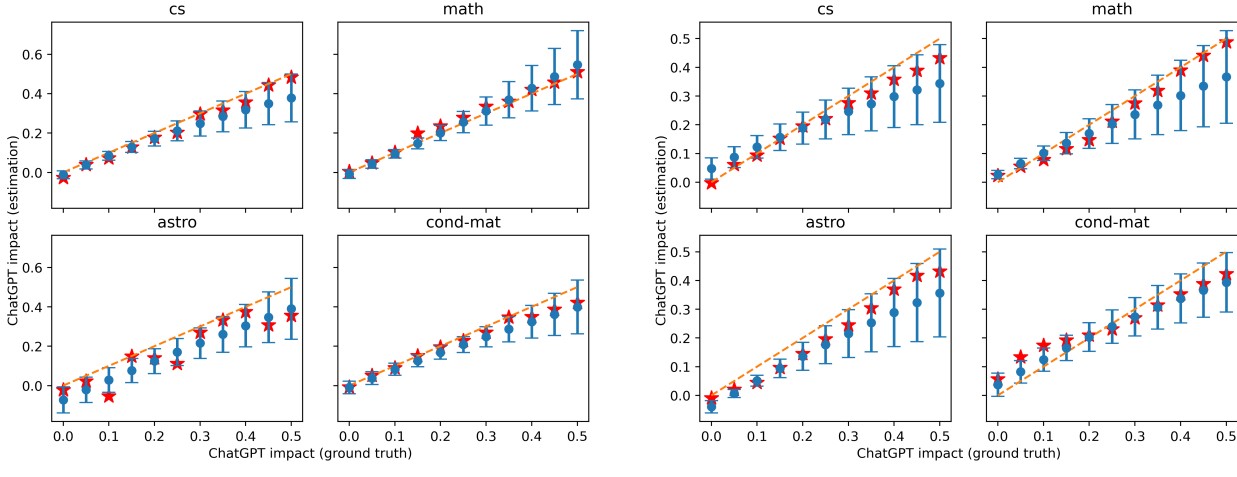

(a) Normalized to the total number of abstracts.

(b) Normalized to the total number of words.

Figure 3: Test results for simulated admixtures of abstracts in period 14. The error bars represent the standard deviation of the estimation results, and the red star is the estimated value of $\eta_n'$ from test data based on optimal $I_j$ with the same mixed ratio $\eta_n$ as in the calibration data. The orange dashed lines correspond to perfect estimation.

## 4.5 Influence of Prompts

Because one may use a wide variety of prompts in practical applications, we also evaluated the robustness of our approach by adopting different prompts for generating the test data than the one we used for calibration. Figure 4 provides an analysis of the effects on prompts, where we also test the following prompt:

**P2**: *"Please rewrite the following paragraph from an academic paper:"*

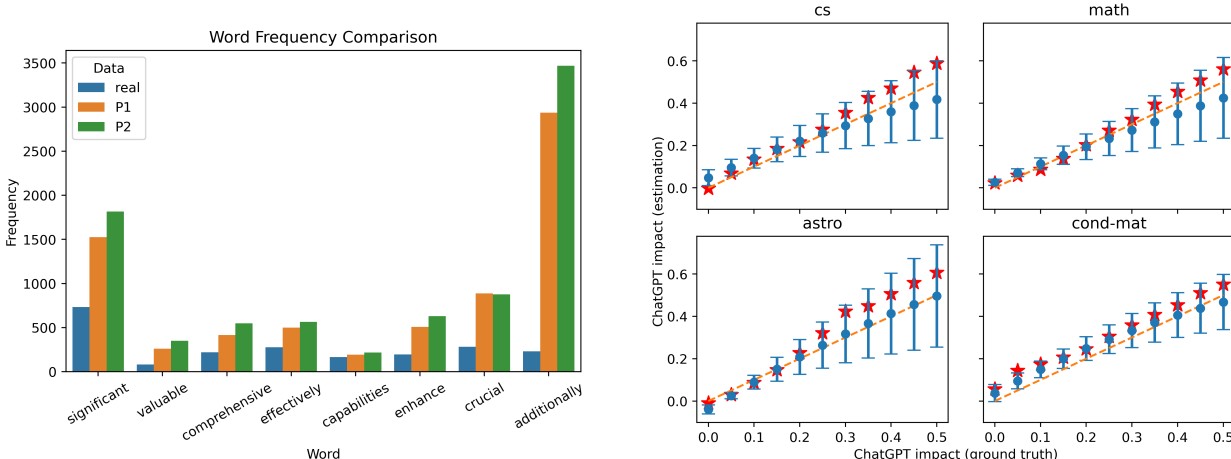

(a) Comparison of word frequency using different prompts. The blue column represents the word frequency in the real abstracts, while the other two columns denote the word frequency in the abstracts processed by ChatGPT using prompts **P1** and **P2**, respectively.

(b) Test results for simulated admixtures of abstracts in period 14. Simulation results with prompt **P2** are used for test data. Other settings are the same as in Figure 3.

Figure 4: Simulation results based on 10000 abstracts in period 14.

Compared to prompt **P1**, we add the word "please" and make it clear that this comes from an "academic paper", replacing "revise" with "rewrite" in prompt **P2**. Figure 4a illustrates the impact of different prompts on the frequency of certain words. Most of these words have a higher frequency when using prompt **P2** compared to **P1**, and much higher than in the real abstracts.

Although the quantitative results in Figure 4b were not as good as before, the errors were still small at lower mixed ratios, hich also illustrates the robustness of our method. This is understandable because in data generated with different prompts, not all of our previous assumptions hold, and the estimate of $\hat{r}_{ij}$ on $r_{ij}$ in our model may be biased. We can also note that most of our estimates here are on the high side relative to the ground truth, most likely because we use a more precise prompt for the test data here, making the frequency change rate of the relevant words higher.

Our estimates are founded on a population level and based on the output of simple prompts. Using more precise prompts, it is entirely possible to achieve abstracts that are more ChatGPT-like (or LLM-like) than our simulations. In addition, in the real world people might use LLMs other than ChatGPT to revise articles, which may have similar but not identical word preferences to ChatGPT, or different noise properties. The above results further reinforce our points: the estimates regarding the usage proportion or impact of LLMs should be treated as relative values under certain conditions, and using numerical expressions without explanation can easily lead to misunderstandings.

### 4.6 Estimation from Real Data

Based on the results in Figure 3 and insights from Eq. (26), we adopted a two-step approach for estimation.

Firstly, we make the first estimate of $\eta_j(t)$, using the 11 word sets $I_j$ for different injected values of $\eta_n$ obtained in our previous calibration results. Then, we can select the word set that corresponds to the mean of the estimated values of $\eta_j(t)$ obtained in the previous step (still from the calibration process) to estimate $\eta_j(t)$ again, with the results shown as the star points in Figure 5a. Alternatively, we can choose three values of $\eta_n$ that are closest to the mean of the first estimation, and use their optimal word set $I_j$ in the calibration procedure for a second round of estimation, leading to the triangle points shown in Figure 5b.

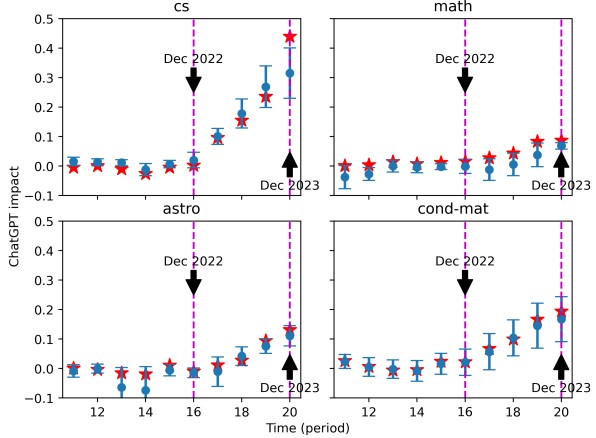
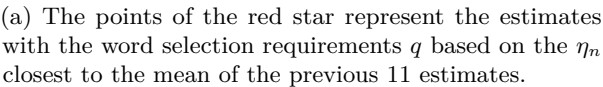
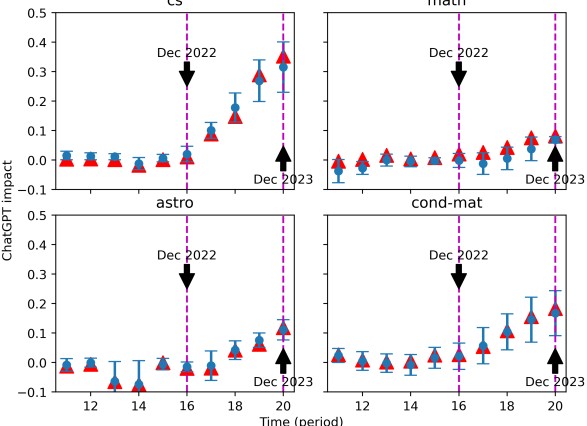

(a) The points of the red star represent the estimates with the word selection requirements $q$ based on the $\eta_n$ closest to the mean of the previous 11 estimates.

(b) The points of triangle represent the average of the 3 estimates, corresponding to the 3 word selection requirements $q$ based on the 3 $\eta_n$ closest to the mean of the previous 11 estimates.

Figure 5: Estimates of $\eta_j(t)$ (i.e., ChatGPT impact) from real data. Word frequencies were normalized on the number of abstracts in each period before the estimation was performed. The error bars represent the standard deviation of the estimation results, using 11 different word sets $I_j$ obtained in the calibration procedure with 11 different $\eta_n$.

Our estimates on $\eta_j(t)$ hover around 0 until 2023, which gives reassurance of the reliability of our methodology. More and more abstracts are being influenced by ChatGPT, especially in the *cs* category, starting from December 2022, after the release of ChatGPT. The results indicates that the density of ChatGPT style texts of the most recent time period in this category is around 35%, when we use the results of one simple prompt, "revise the following sentences", as a baseline. By contrast, we detected a much smaller uptick in ChatGPT impact in *math*, while *astro* and *cond-mat* both reach values between 10% and 20%, approximately.

The ChatGPT impact or LLM impact here is a relative value that corresponds to the change in word frequency from the use of simple prompts. More precise prompts, both in reality and in simulation, could potentially lead to an impact value greater than 1. The presence of the "ChatGPT style" or "LLM style" in a paper does not necessarily mean that the authors directly utilized LLM to generate or modify it. It is also possible that the authors used LLM in another context and that, as a result, their writing habits were influenced by the LLM style – not a remote possibility.

## 5    Discussions

Is ChatGPT transforming academics' writing style? An important question before these discussions is the evaluation of the actual penetration of the usage of ChatGPT in academic writing – without a quantitative estimate, the debate is founded on anecdotal evidence.

As our results have shown, LLMs are having an increasing impact on academic publications. This trend is hard to avoid, and we need to adapt gradually. It is worth considering in this context that reading and writing in English is more difficult for non-native English academics (Amano et al., 2023). With the increasing influx of young researchers, especially non-native English speakers, LLM tools represented by ChatGPT, are transforming academic writing, at least for some disciplines.

Before ChatGPT was released, the pros and cons of other tools were discussed, such as Google Translate (Mundt & Groves, 2016) and Grammarly (Fitria, 2021), but ChatGPT has a much wider range of application scenarios – not to mention, a much higher flexibility. Even if you refuse to use them, your language use is likely to be influenced indirectly by being exposes to their styles via the material you read.

We are more interested in the density of LLM-style texts and its relative value (comparisons between categories and over time) than in establishing how many people are using LLMs – this can be estimated with the help of questionnaires, and it is not possible to get an accurate estimate only based on simulated data.

We have seen similar AI-induced seismic shifts in the past: after AlphaGo (Silver et al., 2017) shocked the world, professional Go players have begun training with AI, and the sport of Go has been profoundly changed as a result (Kang et al., 2022). A similar story may be happening with academic writing, especially for researchers whose first language is not English (Hwang et al., 2023).

## 6    Conclusions

Yes, ChatGPT is transforming academics' writing style, when viewed from the perspective of word frequency.

We have demonstrated here that we can monitor the impact of LLMs in arXiv abstracts by using simple and transparent statistical methods (e.g., word frequencies), an approach that is easily extendable to other subjects and to the complete text of articles, if with additional computational burdens.

We found convincing evidence of a change in word frequency after ChatGPT's release, consistent with predictions obtained from simulating LLM impact from possible users' prompts. The most enthusiastic community (among the four we investigated) in terms of LLM adoption appears to be that of computer scientists, a result that is perhaps unsurprising. Mathematicians, by contrast, are the least keen.

Thanks to our calibration approach, the final estimates are obtained as simple linear regressions, i.e., Eq. (5). These equations tell us which words should be theoretically selected for estimation, which, to our knowledge is a novel result. In addition to focusing on words with increasing frequency, we also included those with decreasing frequency, which are not covered in other papers.

While other papers present similar conclusions, the model and analyses in our work contribute to a deeper understanding and estimation of the impact of LLMs in academic writing. For example, we have theoretically and experimentally demonstrated the importance of selecting appropriate words for estimation, which not only depends on the frequency of the words themselves in the target texts (e.g., types of articles) and the characteristics of LLMs, but also relates to the actual value of LLM impact.

## Limitations

We are aware that our methods can be further improved. For example, our results follow from analyzing a set of words selected based on the value of $q_{ij}^d$ and $\hat{r}_{ij}$. It is actually possible to fine-tune this criterium for a more accurate word selection, which would theoretically give better results, but would be more computationally expensive. Similarly, trying a larger range of prompts should theoretically result in better estimates. And better estimates may be made by more rigorous analysis, such as considering more complex noise terms.

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

## A Related work

Machine-generated text detection has been an active area of research for several years (Bakhtin et al., 2019; Gehrmann et al., 2019), and it has become even more important after ChatGPT appeared (Mitchell et al., 2023; Guo et al., 2023; Chakraborty et al., 2023). Previous studies have shown that ChatGPT has its own linguistic style (AlAfnan & MohdZuki, 2023). Questions have been raised about the reliability of these detectors (Sadasivan et al., 2023). Detection and counter-detection of LLM-generated text soon developed cat-and-mouse games, such as watermarking (Kirchenbauer et al., 2023), paraphrasing (Sadasivan et al., 2023), and the combination of both (Krishna et al., 2024). Besides, researchers also analyzed whether humans have the ability to distinguish between human-written and machine-generated text (Dugan et al., 2023), which is sometimes difficult even for human experts (Casal & Kessler, 2023).

## B Period divisions

Table 2: First and last arXiv paper identifier of 20 periods.

| period | first paper | last paper |
|--------|-------------|------------|
| 1 | 1805.08929 | 1810.00786 |
| 2 | 1810.00787 | 1902.00889 |
| 3 | 1902.00890 | 1905.13537 |
| 4 | 1905.13538 | 1909.11935 |
| 5 | 1909.11936 | 2001.06560 |
| 6 | 2001.06561 | 2005.02178 |
| 7 | 2005.02179 | 2008.04251 |
| 8 | 2008.04252 | 2011.09225 |
| 9 | 2011.09226 | 2103.01828 |
| 10 | 2103.01829 | 2106.04209 |
| 11 | 2106.04210 | 2109.09152 |
| 12 | 2109.09153 | 2112.12197 |
| 13 | 2112.12198 | 2204.01835 |
| 14 | 2204.01836 | 2207.06075 |
| 15 | 2207.06076 | 2210.10618 |
| 16 | 2210.10619 | 2301.10909 |
| 17 | 2301.10910 | 2304.13927 |
| 18 | 2304.13928 | 2307.10978 |
| 19 | 2307.10979 | 2310.09716 |
| 20 | 2310.09717 | 2401.02417 |

## C arXiv Categories

Formally, arXiv has 8 categories in total: physics, mathematics, computer science, quantitative biology, quantitative finance, statistics, electrical engineering and systems science, economics. The first 3 categories contribute the vast majority of arXiv articles, around 91% among the 1 million articles. Hence, we divided the physics papers into sub-categories: astrophysics, condensed matter, high energy physics, etc. The four categories (computer science, mathematics, astrophysics, condensed matter) we selected account for 70% of the total number of articles. To avoid repetition, we also only count the first category of the article for those that have multiple categories (cross-postings).

## D    Other Observations

We define the change factor in the frequency of word $i$, $R_i$, as follows:

$$R_i = \frac{\max_t(f_i(t)) - \min_t(f_i(t))}{\max_t(f_i(t))} \tag{30}$$

where $f_i(t)$ is the count of word $i$ during the time period $t$.

Similarly, we define a change factor in the frequency of word $i$, $R_i'$:

$$R_i' = \frac{\max_t(f_i'(t)) - \min_t(f_i'(t))}{\max_t(f_i'(t))} \tag{31}$$

where $f_i'(t)$ is the count of word $i$ in period $t$, normalized to the same value of $\sum_i f_i(t)$ for all periods $t$.

Figure 6a and Figure 6b illustrate that most of the words with the largest change rate in the time period considered (generally, an increase) in the abstracts are related to hot research topics of the last few years, such as "Covid-19", "LLMs", "AI".

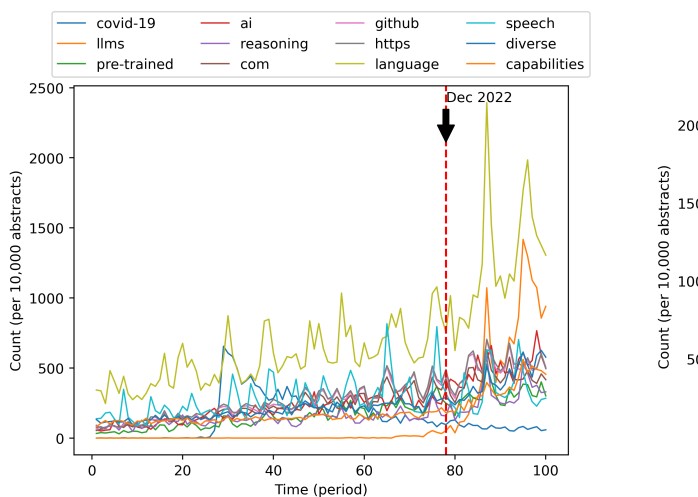
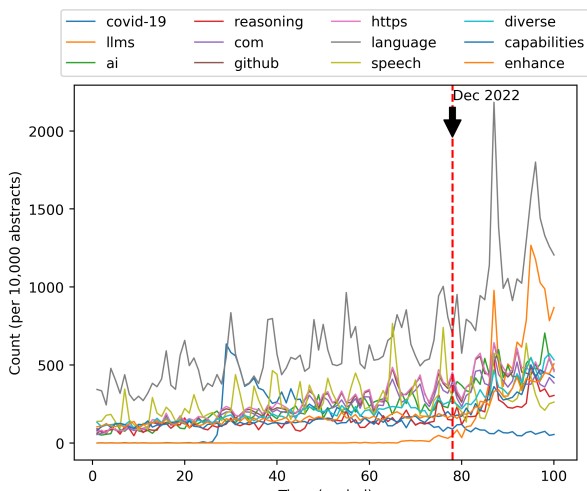

(a) The 12 words with the highest change rate $R_i$ and satisfying $\max_t(f_i(t)) > 500$.

(b) The 12 words with the highest change rate $R_i'$ and satisfying $\max_t(f_i'(t)) > 500$.

Figure 6: Words with the highest change rate in frequency

The total number of words in all abstracts of the first period is used as a base to normalize the frequency of words in the other periods, and the corresponding results are shown Figure 7.

## E    Correlation between Simulated and Real Data

We also defined the word frequency change in all abstracts from year $t-1$ to year $t$, $R_{ij,t}$:

$$R_{ij,t} = \frac{F_{ij,t} - F_{ij,t-1}}{F_{ij,t-1}}, \tag{32}$$

where $F_{ij,t}$ represent frequency of word $i$ per arXiv abstract in category $j$ in year $t$.

Only words with a frequency larger than 0.1 times per abstract before ChatGPT processing are plotted in Figure 8a and Figure 8b. The correlation coefficient between the word frequency change in arXiv abstracts and our estimated ChatGPT-induced word frequency change is very small in all four categories of abstracts, as shown in Figure 8a.

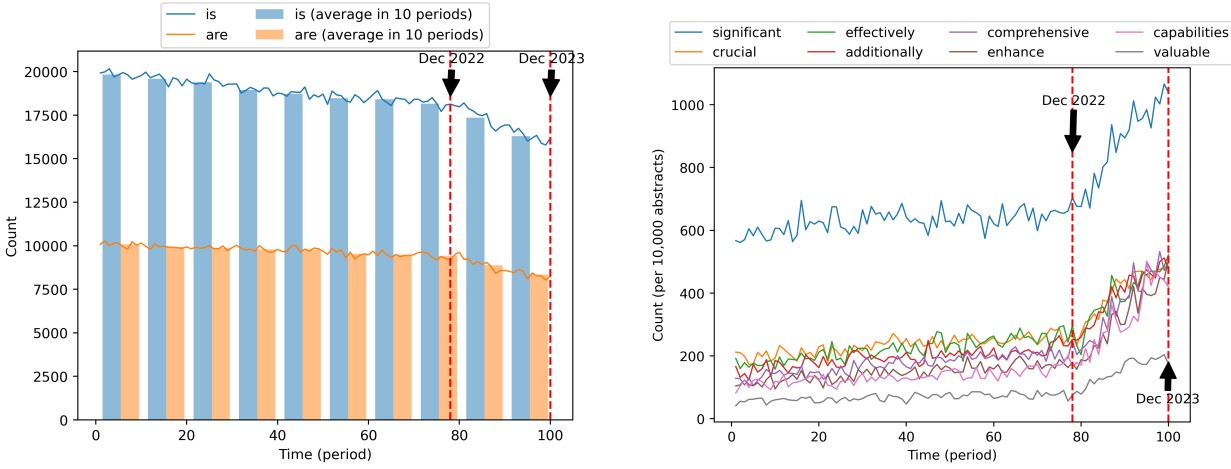

Figure 7: Word frequency changes (with different normalization) in abstracts.

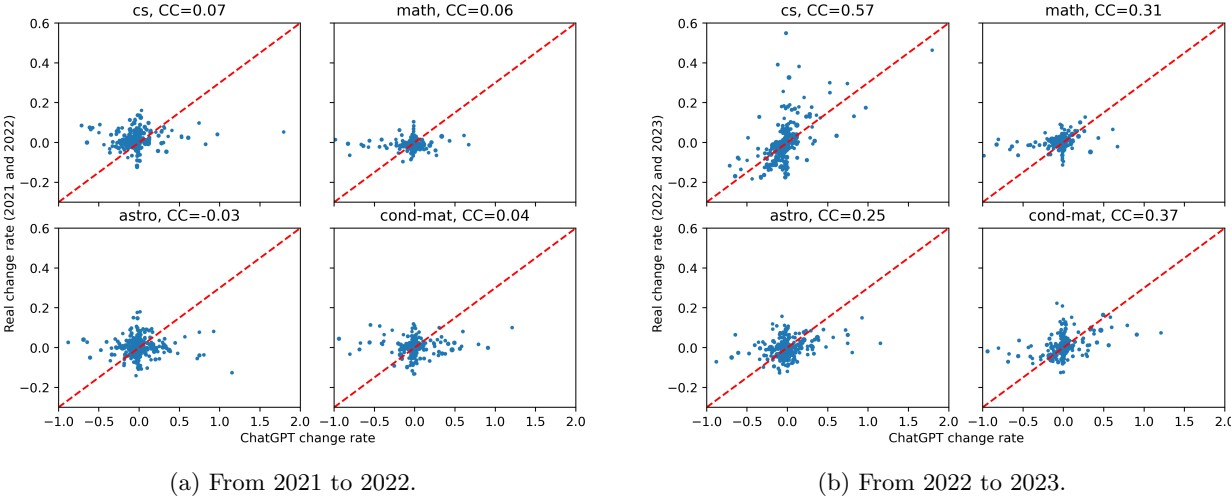

(a) From 2021 to 2022.

(b) From 2022 to 2023.

Figure 8: Comparison of the predicted frequency change rate due to ChatGPT $\hat{r}_{ij}$ (x-axis) and the actual word frequency change for all abstracts (y-axis). CC indicates the correlation coefficient.

However, Figure 8b presents a totally different pattern, where $\hat{r}_{ij}$ and $R_{ij,2023}$ are strongly correlated, especially in computer science abstracts. Although many words seem insensitive to ChatGPT, we can still see a positive correlation for some words in this figure, even among the other categories.

# F Parameters

## F.1 ChatGPT Simulations

- model: gpt-3.5-turbo-1106

- temperature: 0.7

- seed: 1106

- top_p: 0.2

### F.2 Words Selection

- $\frac{1}{q_{ij}^d}$: 10, 20, 30, 40, 50, 60, 70, 80, 100, 150, 200, 500

- $\hat{r}_{ij}$: 0.1, 0.15, 0.2, 0,3, 0.4, 0.5, 0.6, 0.7, 0.8 (corresponding value of $\frac{\hat{r}_{ij} + 1}{\hat{r}_{ij}^2}$)

Below we list the sets of words used to estimate the results in Figure 2.

When we take $\frac{1}{q_{ij}^d} < 10$ and $\frac{\hat{r}_{ij} + 1}{\hat{r}_{ij}^2} < \frac{0.1 + 1}{0.1^2}$ for abstracts in computer science, the words that satisfy the conditions are: 'the', 'is', 'for', 'by', 'be', 'this', 'are', 'i', 'at', 'which', 'an', 'have', 'but', 'we', 'all', 'they', 'one', 'has', 'their', 'other', 'there', 'more', 'new', 'any', 'these', 'time', 'than', 'some', 'only', 'two', 'into', 'them', 'our', 'under', 'first', 'most', 'then', 'over', 'work', 'where', 'many', 'through', 'well', 'how', 'even', 'while', 'however', 'high', 'given', 'present', 'large', 'research', 'different', 'set', 'study', 'important', 'several', 'e', 'further', 'including', 'often', 'provide', 'due', 'using', 'better', 'various', 'problem', 'show', 'problems', 'design', 'proposed', 'g', 'across', 'approach', 'existing', 'compared', 'task', 'learn', 'improve', 'achieve', 'novel', 'domain', 'demonstrate', 'introduce', 'propose', 'prediction'.

And when $\frac{1}{q_{ij}^d} < 50$ and $\frac{\hat{r}_{ij} + 1}{\hat{r}_{ij}^2} < \frac{0.8 + 1}{0.8^2}$, the words are: 'i', 'would', 'so', 'some', 'what', 'out', 'work', 'very', 'because', 'much', 'good', 'way', 'great', 'here', 'since', 'might', 'last', 'end', 'means', 'having', 'thus', 'above', 'give', 'e', 'further', 'far', 'find', 'although', 'show', 'n', 'help', 'together', 'particular', 'whose', 'issue', 'according', 'addition', 'usually', 'art', 'especially', 'respect', 'works', 'shows', 'g', 'makes', 'hard', 'significant', 'run', 'address', 'particularly', 'idea', 'consider', 'includes', 'built', 'adopted', 'obtain', 'establish', 'useful', 'leading', 'performed', 'create', 'named', 'conducted', 'resulting', 'hence', 'findings', 'towards', 'prove', 'build', 'perform', 'moreover', 'describe', 'besides', 'demonstrated', 'via', 'presents', 'mainly', 'fail', 'namely', 'allowing', 'demonstrate', 'advances', 'suffer', 'overcome', 'introduce', 'accurately', 'identifying', 'enhance', 'crucial', 'etc', 'utilize', 'demonstrates', 'additionally', 'focuses', 'motivated', 'characterize'.

## G Noise Analysis

### G.1 Variance in Real Data

Abstracts in the *cs* category among the first 500,000 articles were divided into groups in chronological order, with the same number in each group. We counted the number of occurrences of each word within each group, and calculated the variance between the different groups. This was repeated as a function of the number of abstracts included in each group, and the results are shown in Figure 9a.

Then we also analyzed the coefficient of variation (defined as the standard deviation of the sum divided by the mean of the sum) for the 12 most frequent words, as shown in Figure 9b, and the variance-to-mean ratio (defined as the variance of the sum of a word's counts divided by the mean of the sum), as shown in Figure 10.

We observe that, at least for a subset of the words considered here, the variance-to-mean ratios are essentially on the same scale (although there are words that do not follow this pattern). Therefore, a simple Gaussian distribution

$$\delta_{ij}(f_{ij}) \sim \mathcal{N}(0, f_{ij}\sigma_{ij}^2).\tag{33}$$

which corresponds to case 2, seems to be a reasonable approximation.

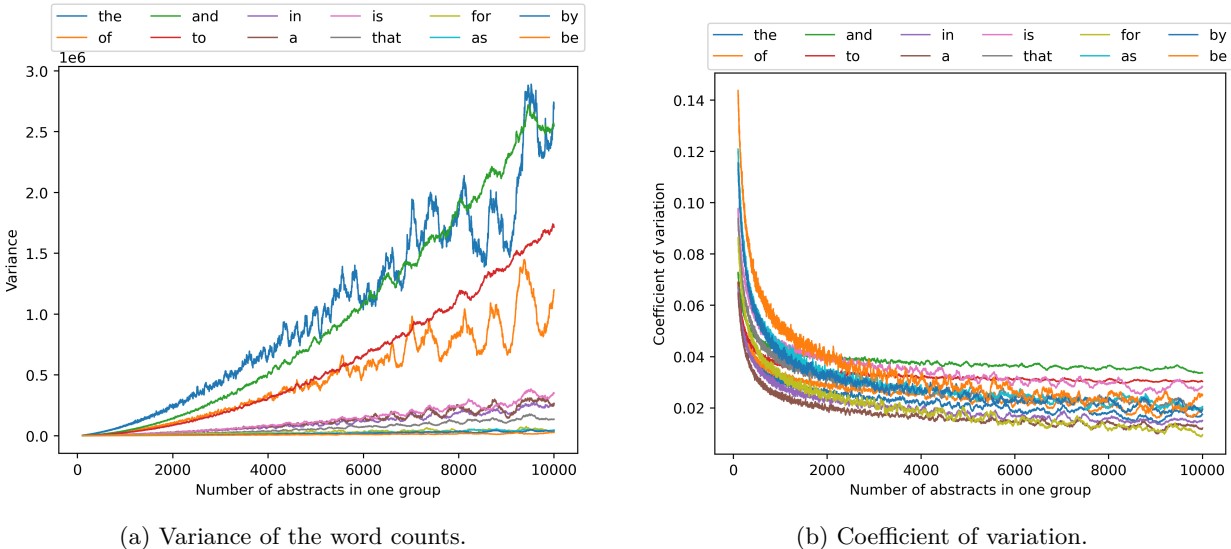

(a) Variance of the word counts.

(b) Coefficient of variation.

Figure 9: Variance of the 12 most frequent words.

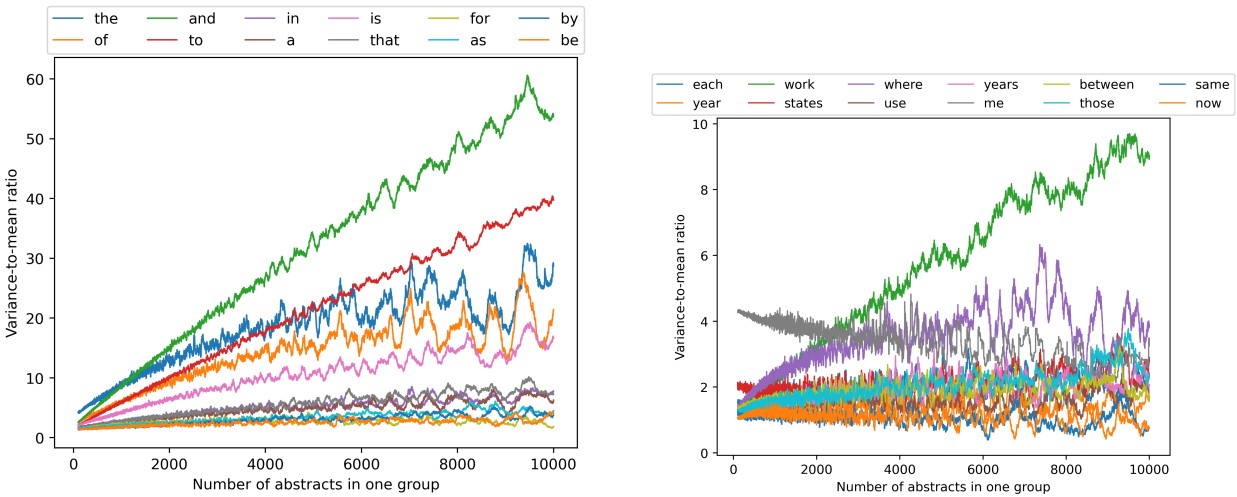

Figure 10: Variance-to-mean ratio

### G.2 Calculation Details

**Case 2:** We can define $g_{ij}^c(t)$ and $\xi_{ij}^c(t)$:

$$g_{ij}^c(t) = \eta_j(t) f_{ij}^*(t) \epsilon_{ij}^\eta(q, f, t) + \sqrt{\eta_j(t) f_{ij}^*(t)} (\hat{r}_{ij} + \epsilon_{ij}^\eta(q, f, t)) \delta_{ij}(1) \tag{34}$$

$$\xi_{ij}^c(t) = \sqrt{f_{ij}^*(t)} \delta_{ij}'(1) \tag{35}$$

As $\xi_{ij}^c(t)$ doesn't depend on $\eta_j(t)$, the loss function under this assumption is:

$$L_{j,t,g}^c(\eta_j) = \frac{1}{n_j} \sum_{i \in I_j} (h_{ij}(t) - \eta_j(t) x_{ij}(t) - g_{ij}^c(t))^2 = \frac{1}{n_j} \sum_{i \in I_j} (\xi_{ij}^c(t))^2. \tag{36}$$

And we will get a complex expression for the bias part like Eq. (22).

As in case 1, we set $\frac{\partial L_{j,t,g}^c(\eta_j)}{\partial \eta_j} = 0$ to obtain the new estimate $\hat{\eta}_j^g(t)$ corrected for bias and noise,

$$(\hat{\eta}_j^g(t) - \hat{\eta}_j(t)) \sum_{i \in I_j} x_{ij}^2(t) = \sum_{i \in I_j} \frac{\partial g_{ij}^c(t)}{\partial \eta_j(t)} (h_{ij}(t) - \eta_j(t)x_{ij}(t))$$
$$- \sum_{i \in I_j} x_{ij}(t)g_{ij}^c(t) - \sum_{i \in I_j} g_{ij}^c(t)\frac{\partial g_{ij}^c(t)}{\partial \eta_j(t)} \tag{37}$$

where

$$\frac{\partial g_{ij}^c(t)}{\partial \eta_j(t)} = f_{ij}^*(t)\epsilon_{ij}^\eta(q,f,t) + \eta_j(t)f_{ij}^*\frac{\partial \epsilon_{ij}^\eta(q,f,t)}{\partial \eta_j(t)} + \frac{\sqrt{f_{ij}^*(t)}}{2\sqrt{\eta_j(t)}}(\hat{r}_{ij} + \epsilon_{ij}^\eta(q,f,t))\delta_{ij}(1)$$
$$+ \sqrt{\eta_j(t)f_{ij}^*(t)}\frac{\partial \epsilon_{ij}^\eta(q,f,t)}{\partial \eta_j(t)}\delta_{ij}(1). \tag{38}$$

The bias part is also expressed as

$$\hat{\eta}_j(t) - \hat{\eta}_j^g(t) = \frac{\sum_{i \in I_j} \mathrm{E}\left[g_{ij}^c(t)\frac{\partial g_{ij}^c(t)}{\partial \eta_j(t)}\right]}{\sum_{i \in I_j}(f_{ij}^*(t)\hat{r}_{ij})^2}. \tag{39}$$

Also with the same assumptions for $\epsilon_{ij}(\cdot)$ and $\epsilon_{ij}^s(\cdot)$, $\epsilon_{ij}(f_{ij}) \sim \mathcal{N}(0, f_{ij}\sigma_{ij,\epsilon}^2)$ and $\epsilon_{ij}^s(f_{ij}) \sim \mathcal{N}(0, f_{ij}\sigma_{ij,\epsilon}^2)$. then we can obtain an expression for $\epsilon_{ij}^\eta(q,f,t)$,

$$\epsilon_{ij}^\eta(q,f,t) = \frac{\epsilon_{ij}(1)}{\sqrt{\eta_j(t)f_{ij}^*(t) + \sqrt{\eta_j(t)f_{ij}^*(t)}\delta_{ij}(1)}} - \frac{\epsilon_{ij}^s(1)}{\sqrt{q_{ij}^d}} \tag{40}$$

and its derivative,

$$\frac{\partial \epsilon_{ij}^\eta(q,f,t)}{\partial \eta_j(t)} = \frac{-\left(2f_{ij}^*(t)\sqrt{\eta_j(t)} + \sqrt{f_{ij}^*(t)}\delta_{ij}(1)\right)\epsilon_{ij}(1)}{4\sqrt{\eta_j(t)}\left(\eta_j(t)f_{ij}^*(t) + \sqrt{\eta_j(t)f_{ij}^*(t)}\delta_{ij}(1)\right)^{\frac{3}{2}}}. \tag{41}$$

Combining the above equations, we can get similar conclusions as in case 1.

