# OpenReview forum: "Is ChatGPT Transforming Academics' Writing Style?"
_TMLR — Rejected by TMLR_

### Review · Reviewer_i5TF · 2025-02-03

**Summary Of Contributions:**

The authors analyze the fraction of arXiv abstracts which have been influenced by ChatGPT since its release. Here, "influenced" does not necessarily mean directly generated or edited by ChatGPT or other LLMs; second-order changes to authors' writing styles are also taken into account. They accomplish this estimation by proposing a model for changes in word frequencies over time; using this model, estimates for the fraction of influenced text can be recovered by solving a least-squares problem. Their framework suggests a method for adaptively selecting a vocabulary of words to consider for the estimation procedure in order to reduce the variance of the estimate. They validate their method on synthetic data and then apply it to real-world abstracts written after the release of ChatGPT. They find that abstracts submitted to cs have seen a substantial increase in ChatGPT influence, while other categories like math have been less affected.

**Audience:**

No

**Claims And Evidence:**

No

**Requested Changes:**

Please address all of the points raised in the Weaknesses section above.

**Strengths And Weaknesses:**

# Strengths

Monitoring LLM usage in academic writing is a timely topic which is relevant to the community. Like previous approaches to this problem, the method introduced by the authors analyzes word frequencies in a corpus of text to estimate the fraction of LLM impact. However, unlike previous approaches, they introduce a method for automatically selecting the word vocabulary which should be used, rather than using heuristic categories like adjectives, etc.

# Weaknesses

The technical derivations are not clear enough to determine whether the derivations are correct or not, and the clarity of the paper overall could be substantially improved. Here are a list of instances of quantities/assumptions which are not explicitly defined in the derivations of Section 4:
- Equation (3): Assumes that $\bar{r}_{ij}$ is constant over different time periods $t$.
- Equation (4): $f_{ij}^d(t)$ is not defined. (It is stated that this is the empirical frequency of the i-th word in j-th category at time t, but explicit instructions for how this is calculated are not given and indeed it is not clear to me what exactly this means.)
- Equations (6) & (7): $\delta_{ij}(\eta_j(t)f^*_{ij}(t))$ and $\delta_{ij}((1-\eta_j(t))f^*_{ij}(t))$ are not defined.
- Equation (8): There is no formula for $C_{ij}$, it is just stated that "the function $C_{ij}(\cdot)$ means the frequency after LLM process".
- Before equation (9): $\epsilon_{ij}$ and $\epsilon^s_{ij}$ are not defined.
- Equation (16): The authors use the definition $h_{ij}(t) = f^d_{ij}(t) - f^*_{ij}(t)$, but based on equation (5) we already have $f^d_{ij} - f^*_{ij} = \delta_{ij}(f^*_{ij})$. Unnecessarily adding notation increases the difficulty of following the derivations.
- Equation (17): It is not clear why the second equality holds.
- After equation (20): $\delta_{ij}(*)$ is stated to be a random variable. What does it mean to take its derivative with respect to $\eta_j(t)$?

There are more examples as the derivations continue, but this list already constitutes an unacceptable level of problems for this manuscript to be published. The technical sections should be thorough reviewed and polished so that a reader can follow it.

The authors mention two "preprints" by Liang et al. which address a similar topic. However, both of these papers have been published well ahead of the submission of this manuscript (in [ICML 2024](https://openreview.net/forum?id=bX3J7ho18S) and [COLM 2024](https://openreview.net/forum?id=YX7QnhxESU#discussion), respectively). Given that the latter paper actually estimates the exact same quantity--the fraction of arXiv abstracts which have been substantially influenced by ChatGPT--the authors should include comparisons with this methodology in their experiments.

It is also worth mentioning that the derivation and final algorithm proposed by this paper are substantially more complicated than the simple two-stage MLE approach used by Liang et al. While TMLR does not require state-of-the-art performance metrics as an acceptance criterion, I believe the authors should provide some justification for why the results of this paper are interesting or useful in light of previous works in order for this to be considered of interest to the community. Such reasons could include, but are not limited to: better validation performance; lower computational cost or required number of LLM queries to produce the training data; or empirical insights not discussed in earlier works.

---

> ### Author Response · Authors · 2025-02-04
>
> Thank you very much for your questions and pointing out some issues. We now provide some quick responses here.
>
> - Equation (3): $\bar{r}_{ij}$ is defined in Equation (2), which represents the word frequency change before and after being processed by ChatGPT, so it's it is independent of $t$.
>
> - Equation (4): $f^d_{ij}$ is defined in Equation (5). We apologize for any misunderstandings caused here and will be sure to correct them in the next version. And in section 5.1, we gave one example, please find the paragraph starting with "We used the first 10 periods before ChatGPT was introduced"
>
> - Equations (6) & (7): $\delta_{ij}(\eta_j(t)f^*_{ij}(t))$ and $\delta_{ij}((1-\eta_j(t)) f^*_{ij}(t))$ are two noise terms of $\delta_{ij}(\cdot)$ with different parameters: $\eta_j(t)f^*_{ij}(t)$ and $(1-\eta_j(t)) f^*_{ij}(t)$. Please also refer to equation (5).
>
> - Equation (8): the formula of $C_{ij}$ is defined in equation (10) below.
>
> - Before equation (9): Here is just a general definition of $\epsilon_{ij}(\cdot)$ and $\epsilon^s_{ij}(\cdot)$. For the exact expression, see equation (24) and the paragraph above it.
>
> - Equation (16):  one is $f^d_{ij}(t) - f^*_{ij}(t)$, and another is $f^d_{ij} - f^*_{ij}$, where the latter does not contain $t$, there is a slight difference.. We apologize for the misunderstanding here and will make the necessary changes in the next version.
>
> - Equation (17): the second equality comes from equation (12) and equation (16).
>
> - After equation (20):  our statement may be easily misunderstood, and we will correct it in the next version.
>
> Liang and his collaborators did outstanding work, especially the one on peer review. And we believe that the fundamental scenarios, such as how to minimize estimation bias, should be considered more carefully before considering a comprehensive analysis, which is the focus of our paper. For example, the alpha estimates are always positive in their COLM article, and at least 2% before ChatGPT, which is clearly a result of having bias. In our paper, LLM impact is around 0 before ChatGPT.
>
> Liang et al. focused on words that are related to LLMs but were not frequently used before, while we chose relatively common words and also considered the words with decreasing frequency. Intuitively, estimation using common words may be more robust. It is also more reasonable to consider words with decreasing frequency.
>
> The formulas in our paper look complicated, but with the help of the calibration, the final estimates are linear regressions (Eq. 18). Most of the other formulas are used to explain how and why certain words are selected for estimation.
>
> If there are any other unclear points, please feel free to continue asking questions.

---

> ### Author Response · Authors · 2025-03-08
>
> Thanks again for your questions and suggestions!
>
> We have revised the paper according to your requests, and the technical derivations should be much clearer. In addition, we have also illustrated the shortcomings of the existing methods as well as our contribution in the introduction and conclusion as suggested by other reviewers, which we believe will be of some interest to the audience. If you have the time to provide further feedback, we would be extremely grateful.

---

### Review · Reviewer_GTTW · 2025-02-06

**Summary Of Contributions:**

(1) An analysis of the arxiv abstracts dataset
(2) A model for (at a population level) the proportion of abstracts that have been influenced by LLMs, with lots of math and some analysis

**Audience:**

Yes

**Claims And Evidence:**

Yes

**Requested Changes:**

I would like you to run some ablations so the reader can understand how crucial all of your modeling choices are. They are:

- Replace the adaptive word selection with simpler alternatives. Here's a few (but feel free to choose your own if you think these are bad): (1) Take top N words by frequency, then select K of those with the largest frequency change between Dec 2022 and Dec 2023. (2) Select a random subset of the top N words. (3) Select the top N words based only on the magnitude of their simulated rij values (the ChatGPT-induced frequency change). This tests whether the calibration and iterative refinement are necessary.

- Remove the second round of estimation (the part where you select the three best word sets and re-estimate), and use only the results from the first estimation based on the initial 11 word sets. This tests whether the iterative refinement improves the accuracy of the estimates or just adds complexity.

- Simplify the noise model. For example: dij(fij) ~ N(0, s^2) - a single variance parameter for all words and categories.

- Try more prompts for both calibration and testing. Include prompts that are much longer with examples (few-shot) - that's what is done in practice! Try prompts that vary along realistic dimensions ("Rewrite this for TMLR audience / NAACL paper" or "Rewrite this to highlight what's novel or impactful" etc).

**Strengths And Weaknesses:**

Thanks to the authors for the hard work on this paper.

Strengths
-------------
- Clear language
- Timely problem
- Everything up to section 4 is easy to understand and is nice to know (but is far from new)

Weaknesses
-----------------
- Section 4 and onwards: this is a very complex model. You justify various criteria throughout but... I don't know if those justifications bear out empirically. And it's hard to know because you don't ablate anything, and because we don't have an unsimulated ground truth. This is a big missing piece, but seems easy-enough to fix. I understand that ablations will have to be w.r.t. simulated gold, as we don't have ground truth here. The ablation requests are specifically described below.

---

> ### Author Response · Authors · 2025-02-09
>
> Thank you very much for your suggestions!
>
> In fact, our initial idea was similar to your suggestion, i.e., using some words for direct estimation, which produced different results depending on the words chosen. As a result, we later developed the calibration approach, which can give us at least the best estimate under some assumptions. In the next version, we will add some results based on simpler methods, to better illustrate and compare the findings.
>
> However, certain aspects cannot be simplified (at least intuitively). For example, different categories of papers use different words, and theoretically, it is better to estimate based on the corresponding changes in word frequency. Noise cannot be simplified, as the noise levels vary across different words (e.g., "is" vs "significant"). The formulas in our paper look complicated, but with the help of the calibration, the final estimates are linear regressions (Eq. 18). Most of the other formulas are used to explain how and why certain words are selected for estimation.
>
> Our current results are based on two prompts, and we will try more prompts, as you suggested.
>
> If there are any other unclear points, please feel free to continue asking questions.

---

### Review · Reviewer_qmFJ · 2025-02-09

**Summary Of Contributions:**

This paper analyzes arXiv papers submitted from May 2018 to January 2024 to investigate the effect of ChatGPT on academic writing. Section 3.1 examines the changes in word frequency distributions in academic papers and reports that words such as "significant" and "effectively" appeared more, and those such as "is" and "our" did less after Dec 2022 (when ChatGPT appeared). Section 3.2 simulates paper rewriting by LLMs by giving the prompt "Revise the following sentences" to GPT-3.5. Section 4 introduces a formalization to consider noises (caused by randomness inside the LLM, uncertainty in word usage, and the epistemic uncertainty in how users actually prompt LLMs). This paper reports that papers in computer science were affected the most by ChatGPT, while other fields, such as mathematics, were not affected so much.

**Audience:**

Yes

**Broader Impact Concerns:**

No concern.

**Claims And Evidence:**

No

**Requested Changes:**

### Major comment

See the weaknesses of this paper. I think the presentation and story of this paper should be improved to highlight the contribution of this work.

### Minor comment

When representing time periods in figures such as Figure 1, please indicate actual dates instead of time intervals.

**Strengths And Weaknesses:**

### Strengths

The research topic is interesting and worth exploring. This study may provide a basis for discussing the effect of automatic research.

### Weaknesses

Section 1 states that the paper analyzes whether the abstract was written by a human or by an LLM. However, the section does not explain the problems of previous research, the differences between previous research and this study, an overview of this study’s approach, the key findings obtained in this study, or their impact. Instead, it cites studies on detecting text generated by LLMs as prior research. However, these studies are quite different from the method used in this paper, making them unsuitable for positioning this research and, conversely, causing the main argument of the paper to become unclear. Additionally, there are references to the pros and cons of analyzing abstracts and the existence of non-native English authors. While this information may be useful, presenting it at a stage where readers do not understand the content of the paper well weakens its effectiveness in positioning the study.

Section 3.1 investigates changes in word frequency in academic papers, reporting that the frequency of words such as "significant" and "effectively" increased after the emergence of ChatGPT in December 2022, while the frequency of words such as "is" and "are" decreased. Section 3.2 simulates text rewriting in academic papers using LLMs by providing GPT-3.5 with the prompt "Revise the following sentences:" However, the details of this simulation are not sufficiently described. For example, it is unclear whether the study focused only on 10,000 abstracts from April 2022 to June 2022 or why GPT-3.5 was used when it aimed to analyze ChatGPT's emergence's impact.

The motivation behind the analysis in Section 4 and beyond is unclear. The introduction to Section 4.1 states that the formulation aims to reduce the impact of prompt variations, but it is not explained how the formulation in Section 4.1 actually achieves this. Additionally, Section 4.2 models noise caused by the randomness of LLM inference and changes in word usage, but the necessity of this approach is unclear. Furthermore, Section 4.3 and later sections describe formulations and derivations based on the least squares method, but the purpose of these formulations is not explained. Finally, the paper does not state whether the content discussed in Section 4 introduces any novelty in analytical methods. As a result, the contribution of Section 4 is not clearly explained in this paper.

In Section 5, the results of calibration are reported. However, it is not explained how the results shown in Figure 2 relate to the paper’s objective of analyzing the impact of ChatGPT on academic writing.

I was expecting to see an answer to the question, "Is ChatGPT Transforming Academics’ Writing Style?" However, this paper does not commit to answering this question. No conclusive evidence supports the idea that ChatGPT changed the writing style.

---

> ### Author Response · Authors · 2025-03-03
>
> Thank you for your constructive questions and suggestions!
>
> We have made many changes in the updated version of the paper that we have uploaded, and we hope they address your concerns.
>
> > Section 1 states that the paper analyzes whether the abstract was written by a human or by an LLM. However, the section does not explain the problems of previous research, the differences between previous research and this study, an overview of this study’s approach, the key findings obtained in this study, or their impact.
>
> Thank you for pointing out these shortcomings. We have made more information in the latter part of the introduction.
>
> >  Instead, it cites studies on detecting text generated by LLMs as prior research. However, these studies are quite different from the method used in this paper, making them unsuitable for positioning this research and, conversely, causing the main argument of the paper to become unclear.
>
> Thanks for your advice. Most of these arguments have been moved into the appendix.
>
> > Additionally, there are references to the pros and cons of analyzing abstracts and the existence of non-native English authors. While this information may be useful, presenting it at a stage where readers do not understand the content of the paper well weakens its effectiveness in positioning the study.
>
> Thanks for your advice. We also moved some contents into the discussion section.
>
> > Section 3.1 investigates changes in word frequency in academic papers, reporting that the frequency of words such as "significant" and "effectively" increased after the emergence of ChatGPT in December 2022, while the frequency of words such as "is" and "are" decreased.
>
> The decrease in frequency is likely due to LLMs avoiding certain terms in their responses.
>
> > Section 3.2 simulates text rewriting in academic papers using LLMs by providing GPT-3.5 with the prompt "Revise the following sentences:" However, the details of this simulation are not sufficiently described. For example, it is unclear whether the study focused only on 10,000 abstracts from April 2022 to June 2022 or why GPT-3.5 was used when it aimed to analyze ChatGPT's emergence's impact.
>
> Thank you for pointing out these gaps. We have added some clarifications in the new version. The results presented in Table 1 are based on the real and simulated abstracts from April 2022 to June 2022, just as an example. In later sections, we consider more abstracts. We used GPT-3.5 for simulations, as it should be the most popular LLM before 2024.
>
> > The motivation behind the analysis in Section 4 and beyond is unclear. The introduction to Section 4.1 states that the formulation aims to reduce the impact of prompt variations, but it is not explained how the formulation in Section 4.1 actually achieves this.
>
> Thanks for your question. We have revised the statement.
>
> > Additionally, Section 4.2 models noise caused by the randomness of LLM inference and changes in word usage, but the necessity of this approach is unclear.
>
> Thanks again for your question. We have added more explanations about the necessity of this approach in the new version, such as Figure 2.
>
> > Furthermore, Section 4.3 and later sections describe formulations and derivations based on the least squares method, but the purpose of these formulations is not explained.
>
> Thank you also for pointing this out. We completely understand your confusion, and this section has been revised extensively. Our aim is to achieve a concise conclusion (so we used OLS), even if the process is a bit more complex.
>
> > Finally, the paper does not state whether the content discussed in Section 4 introduces any novelty in analytical methods. As a result, the contribution of Section 4 is not clearly explained in this paper.
>
> Thank you for your suggestions. We've highlighted our contribution in the new version, as we may have been a little too modest before.
>
> > In Section 5, the results of calibration are reported. However, it is not explained how the results shown in Figure 2 relate to the paper’s objective of analyzing the impact of ChatGPT on academic writing.
>
> Thanks for your question. We have included additional explanations as well. Figure 2 in old version (Figures 3 and 4 in the new verison) illustrates that our method works in the simulated generated test dataset. After this calibration step, we can determine which words to use for the estimation of the final real data, which is also the core of adaptive selection.
>
> > I was expecting to see an answer to the question, "Is ChatGPT Transforming Academics’ Writing Style?" However, this paper does not commit to answering this question. No conclusive evidence supports the idea that ChatGPT changed the writing style.
>
> This is a good question. If we must provide an answer, we think it is "yes", at least in terms of the words used.

---

> > ### Author Response · Authors · 2025-03-03
> >
> > > When representing time periods in figures such as Figure 1, please indicate actual dates instead of time intervals.
> >
> > Thank you for your suggestion. To make the presentation clearer, we have also added the vertical line for December 2023.
> >
> > Thank you once again for your thoughtful review. We are happy to provide further clarifications or make additional revisions if needed.

---

### Author Response · Authors · 2025-03-05

Dear Reviewers,

We deeply appreciate your questions and suggestions, and we hope that the new version of the paper we uploaded a few days ago has addressed your concerns.

We have added some explanations regarding the motivation and process, and adjusted some formulas to make the structure clearer and easier to understand.  The introduction, discussion, and conclusion sections have been rewritten to highlight the main ideas and our contributions.

We are continuously working to improve the paper and would be immensely grateful for any further feedback you could provide.

Thank you once again for your time.

Best,

---

### Decision · Action_Editor_q3bT · 2025-03-18

**Recommendation:** Reject

**Comment:**

My recommendation is based on the fact that the paper is borderline both in terms of its methodological integrity and experimental rigor as well as its relevance to an ML audience.

**Audience:**

Reviewers have noted that the "relevance of this work to the ML community is a bit weak."

The authors could improve the relevance of their work for an ML audience by focusing more on their novel methodological contributions and less on their analysis of how ChatGPT is altering academic writing style, as the latter would be a better fit for an HCI or computational linguistics venue.

**Claims And Evidence:**

The reviewers have pointed out several ways in which the claims made in the paper are unconvincing, and the authors' responses do not adequately address these concerns. Key reasons for rejection are poor notation and too much imprecision in the description of the approach; insufficient comparison to reasonable baselines; and the promise of supporting experiments added to the camera-ready without having been seen by the reviewers.

**Resubmission Of Major Revision:**

The authors may consider submitting a major revision at a later time.